# Ammonia Production Using Bacteria and Yeast toward a Sustainable Society

**DOI:** 10.3390/bioengineering10010082

**Published:** 2023-01-07

**Authors:** Yukio Watanabe, Wataru Aoki, Mitsuyoshi Ueda

**Affiliations:** 1Biotechnology Research Center, Department of Biotechnology, Toyama Prefectural University, Toyama 939-0398, Japan; 2Graduate School of Agriculture, Kyoto University, Kyoto 606-8501, Japan

**Keywords:** ammonia, sustainability, bacteria, yeast

## Abstract

Ammonia is an important chemical that is widely used in fertilizer applications as well as in the steel, chemical, textile, and pharmaceutical industries, which has attracted attention as a potential fuel. Thus, approaches to achieve sustainable ammonia production have attracted considerable attention. In particular, biological approaches are important for achieving a sustainable society because they can produce ammonia under mild conditions with minimal environmental impact compared with chemical methods. For example, nitrogen fixation by nitrogenase in heterogeneous hosts and ammonia production from food waste using microorganisms have been developed. In addition, crop production using nitrogen-fixing bacteria has been considered as a potential approach to achieving a sustainable ammonia economy. This review describes previous research on biological ammonia production and provides insights into achieving a sustainable society.

## 1. Introduction

In recent years, rapid population growth and industrial development have increased the use of several fossil fuels and have increased the amount of waste and environmental pollutants, leading to destruction of the global environment and causing global warming, ocean acidification, and so on [1]. In alleviating this situation, researchers, governments, companies, and other organizations are greatly interested in improving the global environment, and the Sustainable Development Goals have been set forth [2,3]. Fossil fuels are the most significant energy source in the modern world, which are converted to electrical energy primarily through thermal power generation. Various natural energy sources have been used as alternatives to fossil fuels, including wind, hydro, solar, and geothermal energy [4,5,6]. In addition to these renewables, biomass as fuel has also received increasing attention. Biomass includes perennial plants, forestry waste, algae, and food waste (municipal solid waste), which are also recognized as carbon-neutral fuels. However, the supply of biomass is greatly affected by weather, time, location, and economics, making it unstable [7,8,9,10,11,12,13]. In addition, biodegradable materials could be used as sources of biofuels, and ammonia is expected to be used as a biofuel (Figure 1) [14].

### 1.1. Industrial Uses and the Need for Ammonia

Ammonia is an important compound in a variety of industries [15]. Fixed nitrogen, such as ammonia, is essential for crop growth, and increasing the amount of nitrogen circulating on the planet allows for population growth [16]. The production of ammonia is highly important to sustain life, and about 80% of ammonia is used in the production of fertilizers. Ammonia is also used for refrigerant gas or the synthesis of various chemicals such as plastics, explosives (trinitrotoluene, nitroglycerin, and nitrocellulose), textiles (rayon and nylon), agricultural chemicals, dyes (for cotton, wool, silk, etc.), and so on [17,18,19]. In recent years, ammonia-fueled batteries have also been devised [18,20,21,22,23,24].

The potential application of ammonia as fuel has received increasing attention [8,25,26,27,28,29]. Hydrogen has low transport efficiency because of its low volumetric energy density (3 W h·L^−1^) and higher calorific value per liquid wight (141.9 MJ/kg) [30]. Therefore, approaches to converting hydrogen to a more transport-efficient substance have been investigated. Ammonia has a flammable range of 16% to 25% (*v*/*v*), which can be transported more safely than hydrogen [31,32]. Liquid ammonia contains hydrogen atoms per volume and energy density (MJ L^−1^) that are 1.7 and 1.5 times higher than that of liquid hydrogen, respectively [33,34]. Hydrogen has a low boiling point of −253 °C, and it requires considerable energy to liquefy. By contrast, ammonia has a boiling point of −33.4 °C, and it can be easily liquefied by using general-purpose refrigeration equipment and can be handled easily. Thus, it has been considered for use as a carrier for hydrogen in a hydrogen society [32,35,36,37]. In addition, the hurdle to the industrial application of ammonia is lower than that of hydrogen because the infrastructure for storage and transportation of ammonia has already been established.

### 1.2. Chemical Method for Ammonia Production

The Haber–Bosch process is a typical example of ammonia nitrogen fixation, and 55% of the world’s ammonia is produced by this method [38]. This method requires the cleavage of the triple bond of the nitrogen molecule, which uses a large amount of energy [15,39]. The energy used by the Haber–Bosch process is equivalent to 2–3% of the world’s annual fossil fuel use, and it accounts for 1.4% of the global annual carbon dioxide emissions [38,40]. Carbon dioxide is a well-known greenhouse gas (GHG), and its reduction is strongly desired because of environmental issues of growing concern in recent years. Therefore, the Haber–Bosch process has been improved upon in recent years, particularly in catalyst improvements. Moreover, the development of catalysts that allow reactions to proceed under conditions closer to ambient temperature and pressure has been considered. Compared with iron catalysts, Ru-based catalysts enable the fixation of ammonia at lower pressure (90 atm), and they have about 20 times higher catalytic efficiency [41]. However, ruthenium has become increasingly expensive in the last decade, thereby hindering its industrial use [42,43,44]. The performance of catalysts has been greatly studied and further developed, including the development of several molybdenum-based catalysts that mimic the active center of nitrogenase (a nitrogen-fixing enzyme) for the synthesis of ammonia at ambient temperature and for pressure in microorganisms [45,46].

At present, most of the hydrogen required by the Haber–Bosch process is obtained by electrolysis. In general, the operation of a water electrolysis unit requires a continuous supply of highly purified water. Furthermore, 9 tons of highly purified water is required to produce 1 ton of hydrogen. Based on these data, 233.6 million tons of water per year is required to produce 1 ton of ammonia using hydrogen obtained from water electrolysis [47]. With the progression of global warming, improving the global environment is necessary [2]. Therefore, attempts are being made to supply ammonia in a sustainable manner using natural energy (green energy) such as wind and solar power generation [48,49]. Ammonia using hydrogen produced from green energy is known as “green ammonia” [50].

In addition to wood biomass, the amount of food waste has continued to grow excessively in recent years, particularly in developed countries [51,52]. Food waste produces considerable amounts of GHGs and environmental pollutants through landfilling and incineration [53]. Thus, many attempts have been made to obtain energy from food waste [13]. In particular, okara is an abundant food waste, and its various uses are being considered [54,55]. Attempts to produce ammonia from food waste by physicochemical methods have also been studied [56,57,58]. For example, considering that glutamic acid is a common source of nitrogen in food wastes, researchers have targeted glutamic acid contained in sewage sludge and meat and bone meal, and they succeeded in producing ammonia with 35–51% efficiency at 800 °C and 0.5–1.0 g g^−1^ carbon content by automated gasification [56]. This reaction is dependent on the catalytic metal ions in the food waste, which may need to be optimized for each feedstock. The addition of LaFeO_3_ as a catalyst resulted in an efficiency of 54 vol% [57]. Therefore, this approach could potentially produce 10% of the ammonia used in Europe [56].

Several studies have developed sustainable methods to produce ammonia, most of which include green power generation methods, chemical synthesis methods, and sustainable ammonia production using biological methods [59,60,61]. Various methods of recovering ammonia from food wastes have also been used, and research is progressing toward practical application [62]. Alternatively, we focus on sustainable ammonia production methods such as biological methods—bacterial and yeast methods—and explore their application potential (Figure 2).

## 2. Engineered Bacterial Method for Ammonia Production

Microorganisms can degrade and synthesize a wide variety of compounds by applying engineering methods such as directed evolution or genome editing [63,64,65,66]. Based on previous reports, pharmaceuticals that cannot be digested by the human body are broken down by microorganisms in the environment [67]. Considering their capability to degrade and synthesize not only natural substances but also man-made substances, microorganism have been used to synthesize a variety of substances (Figure 3) [68,69].

### 2.1. Bioengineering of Nitrogen Fixation and Metabolic Engineering for Ammonia Production

Compared with the Haber–Bosch process, the use of nitrogenase is highly sustainable because it can produce ammonia at room temperature and pressure [70]. The amount of nitrogen fixed by forage legumes accounts for 17.2 × 10^7^ tons of nitrogen per year [71,72,73]. Thus, approximately 21% of the nitrogen is produced by nitrogen fixation [74].

Two main species of microorganisms possess nitrogenases. One species is the genus *Rhizobia*, which parasitizes legumes, and the other species is the genera *Azotobacter* and *Klebsiella*, which can grow without parasitizing plants and are the main focus of research [75,76]. Nitrogenase utilizes considerable energy, requiring eight molecules of ATP to produce one molecule of ammonia, and it is easily deactivated by oxygen exposure [77]. Nitrogenase is composed of multiple subunits; the roles of each subunit are interrelated, and the molecular mechanism is complex. Thus, the reaction mechanism remains unclear [78,79].

Attempts to produce ammonia using nitrogenase have been pursued, starting with the construction of a heterologous expression system for the nitrogenase subunits from *Klebsiella oxytoca* in *E. coli* [80]. This nitrogenase uses molybdenum as a cofactor, but *Azotobacter vinelandii* exists as a nitrogen-fixing bacterium with a nitrogenase that uses vanadium and iron. Various attempts have already been made with some success to construct heterologous expression of this nitrogenase gene cluster in *E. coli* and yeast, to elucidate the mechanism by which it is protected from oxygen, and to increase its activity [81,82,83]. For example, heterologous expression of the nitrogenase complex of *A. vinelandii* has been successful in *E. coli*, and the minimum number of genes required for nitrogen fixation has been elucidated. *A. vinelandii* possesses an Fe nitrogenase complex, which consists of 21 subunits, and this nitrogenase complex is actively expressed in only 10 genes in *E. coli* [84]. Furthermore, *Paenibacillus* sp. is composed of 20 subunits of the Fe-type nitrogenase and functions with a set of nine genes. However, the activity of the heterologously expressed nitrogenase was approximately 10% compared to the nitrogenase of the wild type [85]. Further studies have shown that adding *Klebsiella oxytoca* NifSU (Fe-S cluster assembly) and *Paenibacillus* electron transporter genes (*pfoABfldA*) to this minimal gene set improved the expression, yielding 50.1% of the activity of the wild type [85]. A further attempt was made to reconstitute them by combining fourteen nitrogenase-related genes from *K. oxytoca* into five gene cassettes and cleaving the subunits with a protease from the tobacco etch virus. As a result, this *E. coli* could grow only nitrogen molecules in the air without oxygen [84].

Various theories have been proposed to investigate the mechanism through which nitrogen-fixing bacteria fix nitrogen without inactivating nitrogenase under aerobic conditions; *Pseudomonas stutzeri* A1501 forms a cyst made of a polysaccharide membrane to protect oxygen, thereby protecting nitrogenase from oxygen [86]. *A. vinelandii* also produces polysaccharides to protect nitrogenase from oxygen by localizing the polysaccharides to the plasma membrane [83].

Apart from the aforementioned methods that can directly improve ammonia production capacity, attempts have been made to increase the ammonia supplied to plants by enhancing the ability of nitrogen-fixing bacteria to bind to plants, which has reached practical application [87]. For example, by ingesting *Pseudomonas protegens* Pf-5 X940, 15 nitrogen isotope dilution analyses showed that corn and wheat produced significant amounts of fixed nitrogen by this organism. In addition, the colony in wheat plants was formed on the root surface of Pf-5 X940 expressing GFP [88].

Recently, for industrial applications, various companies such as Ginkgo Bioworks and Pivot Bio, Inc., have been studying the potential use of nitrogen-fixing microbes. Pivot Bio, Inc., performed a study that used nitrogen-fixing bacterium of the genus *Enterobacter* sp. with genetic engineering [89]. These bacteria dis not have enough glutamine because of the low level of expression of the transcription factor GlnR. Thus, when increasing intracellular glutamine, these bacteria synthesized ammonia in the presence of more nitrogenase than the wild type [89]. In addition, corn (*Zea mays*) seeds with these bacteria were pre-infected and provided a steady nutrient supply [89]. In 2019, the average corn N uptake was higher when inoculated with Proven™ than that without nitrogen fertilizer application, with yields of 0 kg N ha^−^¹ (treated with fertilizer) and 10.9 kg N ha^−^¹ yields (added with bacteria), but this difference was not statistically significant [90].

Many other attempts have been made to use anaerobic microflora to produce ammonia [13,61,91]. A number of methodologies for ammonia recovery have also been proposed, including evaporating the solution after fermentation and increasing the pH [92,93].

### 2.2. Ammonia Production from Food Waste by Using Bioengineering Methods

Several methods have been reported for producing ammonia from food waste and other biomass using engineered *E. coli* and *Bacillus* [54,94,95,96,97]. For example, *Bacillus subtilis* is used to hydrolyze protein biomass with its own protease and to efficiently produce amino acids, and ammonia and alcohol are highly produced from amino acids using metabolic engineering methods [94]. In particular, the authors knocked out the *codY* gene, a regulator of branched-chain amino acids in the *B. subtilis* metabolic pathway. Furthermore, the authors knocked out *bkdB*, a lipoamidoacyltransferase, which inhibits the conversion of 2-keto acids to acyl CoA. Then, 2-keto acids are decarboxylated by heterologously expressed 2-keto acid decarboxylase to provide biofuel accumulation. Furthermore, the overexpression of the alpha-ketoisovalerate decarboxylase gene *leuDH* accelerated ammonia production. Consequently, the biofuel production, including ammonia by deamination from media containing amino acid, produced 46.6% of a theoretical yield [94]. Ammonia production from amino acids using *E. coli* has also been attempted [96,97]. Considering the strong ammonia assimilation capacity to keep the cell in homeostasis (about 20% of nitrogen compounds is stored in the cell) of *E. coli*, attempts were made to knock out genes involved in ammonia assimilation. For example, knockout of the glutamine synthetase *glnA* was predicted to increase the production of ammonia. Furthermore, overexpression of the decarboxylase gene *kivD* was used to bias the equilibrium reaction between ammonia and amino acid production toward the ammonia-producing side. As a result, the production of ammonia from a medium containing amino acids succeeded with a theoretical yield of 47.8% [97]. However, the abovementioned studies were primarily experiments using amino acid–containing media and not actual food wastes. Therefore, researchers investigated ammonia production from six different culture media and four different food wastes using *E. coli* to promote the use of actual food wastes [96]. In addition, metabolic profiling showed a correlation between the concentration of substances such as sugars, organic acids and amino acids in the medium, and ammonia production suggested that glucose inhibits the production of ammonia. The M9 yeast extract medium containing sugar at various concentrations showed a negative correlation with ammonia production. Therefore, researchers knocked out the glucose transporter gene *ptsG*, which transports major sugars such as glucose in *E. coli*, deduced the expression of phosphoenolpyruvate- and phosphotransferase-related genes, and succeeded in producing ammonia from the medium containing amino acid and sugar with about 73% yield. Furthermore, ammonia was successfully produced from pretreated soybean residue with a conversion efficiency of about 47% and a high ammonia concentration of approximately 35 mM [96]. In these studies, ammonia was produced in the cells, indicating a trade-off between microbial growth and ammonia production. Moreover, a study using metabolically modified *B. subtilis* for ammonia production reached a plateau at day 6, and it did not increase at day 7 [95]. These results indicated that the produced ammonia was used for cell growth. Therefore, extracellular ammonia production can simultaneously improve production and growth.

## 3. Engineered Yeast Method for Ammonia Production

In addition to *E. coli*, many attempts have been made to produce ammonia using yeast [98,99]. Numerous attempts have also been made to investigate the heterologous expression of nitrogenases in yeast. For example, the nitrogenase subunit NifB is required for the formation of MoFe clusters at the nitrogenase active center; however, this protein is insoluble in mitochondria of budding yeast [100]. Two combinatorial libraries were constructed for optimization. One library consists of six subclusters including nifUSX and fdxN, and the other library includes 28 NifB genes extracted from the public database and showed different levels of expression based on a factor design [101]. Consequently, NifB genes derived from the archaea *Methanocaldococcus infernus* or *Methanothermobacter thermautotrophicus* were activated in the mitochondria of yeast [101]. Nitrogenase subunits derived from *K. oxytoca* within plant mitochondria were also used to attempt expression. In addition, NifF, NifM, NifN, NifS, NifU, NifW, NifX, NifY, and NifZ could be expressed in a soluble form, whereas NifB, NifE, NifH, NifJ, NifK, NifQ, and NifV are insoluble. However, the NifM activity decreased by 10% compared with the previous study because of N-terminal processing after transportation to the mitochondria [102,103]. Furthermore, a NifD protein derived from *K. oxytoca* contains a mitochondrial targeting peptide (MTP) recognition sequence that undergoes processing, making functional expression difficult in mitochondria, and R98 has been identified as an important residue of this cleavage. As a result, R98P shows to be resistant to processing, and high levels of activity are retained in the mitochondria [104]. Another study found that the Y100Q mutant is resistant to processing within plant and yeast mitochondria [105]. In their search for efficient NifH expression, 32 different NifHs were expressed in tobacco and yeast mitochondria and it was found that NifH from the thermophilic bacterium *Hydrogenobacter thermophilus* is the most active form [106]. Thus, the expression of nitrogenase subunits from thermophilic bacteria is a potential approach that is worthy of further study because the temperature in mitochondria could reach 50 °C [106]. Genetic diversity can be used to identify the most appropriate Nif protein components from large sequence pools to manipulate eukaryotic nitrogenases [107]. Cloning techniques such as codon optimization of gene synthesis and other synthetic biology tools such as metabolic engineering methods allow for the construction of multi-protein pathways containing the proteins’ various sorts of origins. For example, the NifH protein obtained from *H. thermophilus* was found to be soluble in the mitochondria of *Saccharomyces cerevisiae* and *Nicotiana benthamiana*, which accumulate at higher levels than *A. vinelandii* [107]. Therefore, functional nitrogenases in plant cells could be constructed in the future to create practical crops that can fix nitrogen on their own [108]. Recently, researchers have also performed heterologous expression of an active nitrogenase in the chloroplasts of the cyanobacterium *Synechococcus elongatus* PCC7942, creating an algae that can fix nitrogen using energy from photosynthesis [109].

Metabolically engineered yeast can produce large amounts of ammonia because its of exposure to high concentrations of ammonia, thereby causing growth defects [110]. Yeast-based methods for producing ammonia from food waste have primarily used cell surface engineering (Figure 4) [111]. The nitrogen source of food waste is primarily amino acids derived from proteins; thus, amino acid oxidases, which can efficiently produce ammonia from amino acids, are considered suitable for presentation to the cell surface and are efficient in producing ammonia. Several studies have used yeast cell surface display systems [111]. For example, target proteins are presented on the cell surface by adding a signal for secretion to the N-terminus and a cell wall anchor protein including a glycosylphosphatidylinositol anchor attachment signal sequence to the C-terminus. Approximately 10^5^–10^6^ proteins could be presented on the yeast cell surface, and yeast cells can be manipulated as a whole-cell biocatalyst [111]. Furthermore, proteins immobilized on the cell surface can stabilize and exhibit higher enzymatic activity than those in the free state. In addition, the folding machinery of the eukaryote displays a variety of proteins [111]. Thus, given these advantages, the yeast cell surface has been successfully manipulated to produce ethanol from carbohydrates, woody biomass, and large algae with high efficiency [112,113]. Ammonia produced from soybean residues has been used in yeast cell surface engineering. Amino acid catabolic enzymes such as deaminase, transaminase, oxidase, and ammonia lyase are known to produce ammonia from amino acids. Ammonia lyases can be displayed on the yeast cell surface because they do not require any cofactors to make holoenzymes. In a study, glutamine ammonia lyase (YbaS) was displayed on the cell surface, and this yeast successfully produced ammonia from a glutamine solution with high efficiency (83.2%) and high concentration (3.34 g/L) [98]. Moreover, although 0.1% (*v*/*v*) ammonia-containing medium inhibited yeast growth, no impairment of the function of this whole-cell catalyst was observed [98,99]. Furthermore, YbaS-displaying yeast successfully produced ammonia with high efficiency from a solution of enzymatically pretreated okara (soybean residues). However, when producing ammonia using YbaS ammonia lyase, only glutamine was used to produce ammonia among the 20 amino acids [114]. l-amino acid oxidase is known to have broad substrate specificity, and it can produce ammonia from a wide variety of amino acids [99]. For example, a l-amino acid oxidase from *Aplysia californica* is active against l-arginine and l-lysine. In addition, a l-amino acid oxidase obtained from snakes such as *Bothrops atrox*, *Crotalus viridis helleri*, and *Daboia russelii* has been well studied, and it is inactive against amino acids such as l-glutamine and l-aspartic acid [115]. However, it is active against hydrophobic amino acids such as l-leucine and hydrophilic amino acids such as l-histidine and l-lysine. It is also inactive against aromatic amino acids such as l-tyrosine and l-phenylalanine. *Hebeloma cylindrosporum* also contains enzymes with broad specificity, as it has mycelia for the intracellular absorption of ammonia from amino acids contained in soil. In the literature, an enzyme, namely, HcLAAO (l-amino acid oxidase from *H. cylindrosporum*), can produce ammonia from more than 10 known amino acids [116]. Therefore, ammonia can be efficiently produced from food waste using yeast displaying HcLAAO. The display of HcLAAO in budding yeast led to high ammonia production efficiency (about 88%) under mild conditions (30 °C) from a pretreated okara solution [99]. In this study, no toxic effects of extracellular ammonia on catalytic activity were observed, which is consistent with previous studies that used glutamine ammonia lyase. Although these attempts to produce ammonia from food waste have been successful, these studies are all laboratory-scale studies. Enzymes have also been developed as tools to produce ammonia from amino acids, with the recent creation of enzymes that can produce ammonia from 13 amino acids [117]. Therefore, industrial-scale attempts for social implementation should be considered in the future.

## 4. Conclusions

This review mainly focused on ammonia production using genetic engineering methods (Table 1). Energy production with low environmental burden is considered essential for the formation of a sustainable society. In addition to studies in this review, for example, anaerobic culture is also one of the methods currently in practical use for energy production from food wastes [118]. These anaerobic cultures usually do not use genetic engineering methods, such that this method has room for optimization by using genetic engineering introduced in this review.

To achieve a sustainable society, we have to solve some problems other than the development of biological methods such as the storage of food waste [119,120]. In addition, consumption of ammonia is important for creating sustainable environment. For example, ammonia is only 30–50% utilized for crop growth [121]. The remainder eutrophicates the oceans, contaminates drinking water, and, in some areas, changes the composition of vegetation and encourages the propagation of non-native species [122,123,124,125]. Ammonia is converted to N_2_O by microorganisms in the environment, which is also a well-known greenhouse gas, and its emission is also a problem [74,126].

Microorganisms such as *E. coli* and yeast contain nitrogen in their bodies, with about 20% of the total amount as proteins [127]. The nitrogen in cells comes from assimilation of amino acids or proteins in a medium. Therefore, the amount of produced nitrogen must be larger than the input nitrogen for effective biofuel production. To solve this problem, cell surface engineered yeast could be repeatedly used for production because the enzyme on the yeast cell surface is stable [128]. Recently, some studies attempted to estimate the metabolism of microorganisms so that it might be possible to calculate the potential of microorganisms to efficiently produce ammonia in the future [129,130,131].

In this review, we introduced various attempts to improve the Haber–Bosch process, one of the sources of carbon dioxide generation, which uses the largest amount of fossil fuels, in order to realize a sustainable society.

The use of genetically optimized microorganisms to produce ammonia instead of the Haber–Bosch process is a potential and environmentally friendly approach to solve the global ammonia demand problem and to develop a sustainable carbon-free society (Table 2).

Therefore, in the future, it may be possible to produce ammonia more efficiently by using it concurrently with such methodologies.

## Figures and Tables

**Figure 1 bioengineering-10-00082-f001:**
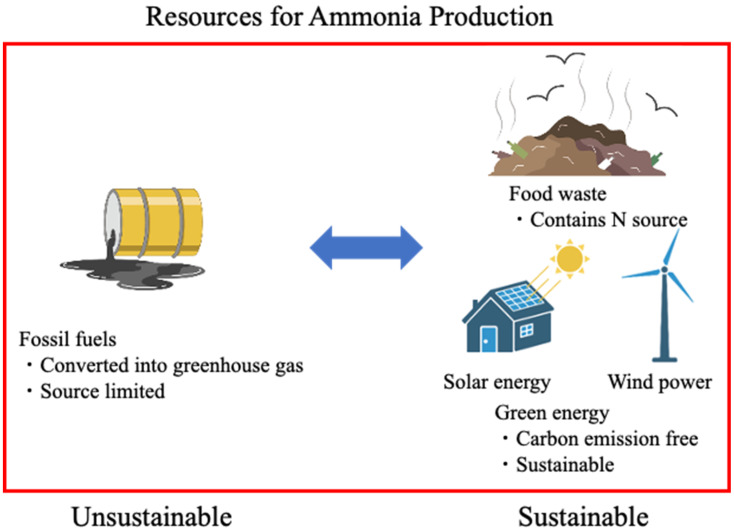
Sustainable resources of energy for a sustainable society (created with Biorender.com).

**Figure 2 bioengineering-10-00082-f002:**
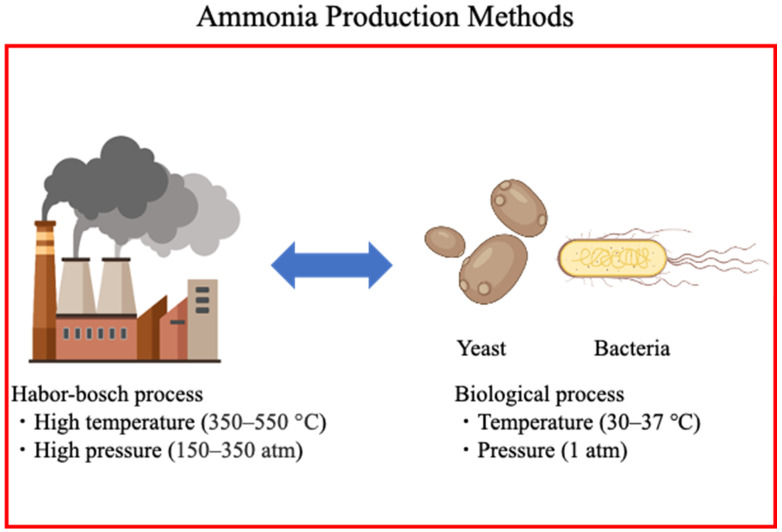
Sustainable ammonia production (created with Biorender.com).

**Figure 3 bioengineering-10-00082-f003:**
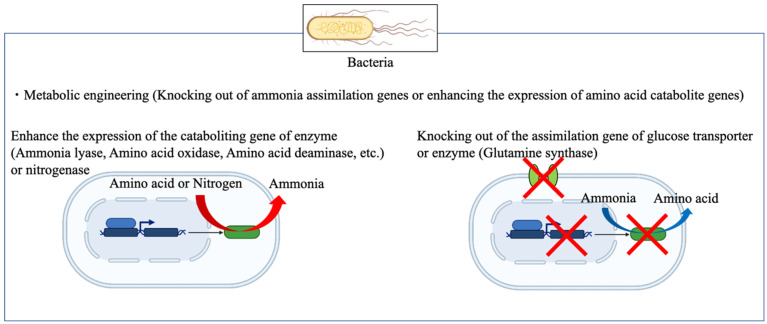
Ammonia production using engineered bacteria (created with Biorender.com).

**Figure 4 bioengineering-10-00082-f004:**
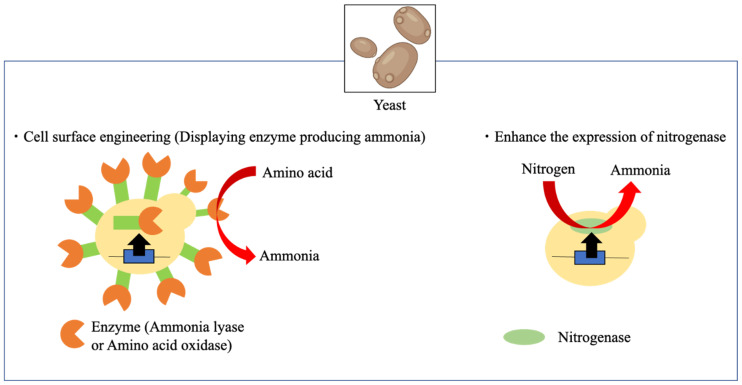
Ammonia production using engineered yeast (created with Biorender.com).

**Table 1 bioengineering-10-00082-t001:** Summary of biological methods for ammonia production.

	Metabolic Engineering	Cell Surface Engineering
Reaction Place	Inside of the cells	Outside of the cells
pH of reaction environment	Neutral condition (pH 7.2–7.8)	Depends on the reaction mixture
Effect on the enzyme	Nothing	Stable (fixed on the cell surface)

**Table 2 bioengineering-10-00082-t002:** Summary of ammonia production methods.

	Chemical Method	Bacterial Method	Yeast Method
Resource	Nitrogen in air Hydrogen	Nitrogen in air Food waste	Food waste
Energy resource	Fossil fuel (coal, oil, or natural gas)	Not required	Not required
ReactionEnvironment	High pressure (150–350 atm) High temperature (350–350 °C)	Pressure (1 atm) Temperature (37 °C)	Pressure (1 atm) Temperature (37 °C)
Point of improvement for sustainable production	Alternative energy resource (solar, wind, hydro, etc.) Novel catalyst to react with ambient pressure and temperature	Heterologous expression of nitrogenase Metabolic engineering to improve ammonia production from amino acids contained in food wastes	Cell engineering (Cell surface engineering) to improve the production of ammonia from food wastes

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
