# Peer review of "Ammonia Production Using Bacteria and Yeast toward a Sustainable Society"

_bioengineering, 2023, doi:10.3390/bioengineering10010082_

Round 1

Reviewer 1 Report

1.      The introduction is too brief. If the reader only reads the introduction, it is hard to think of this as a review of ammonia.

2.      The introduction should show the previous reviews on ammonia production, and explain the difference between this paper and the previous reviews and the novelty of this paper.

3.      Biomass is unstable. Is it feasible to use biomass such as food waste to produce ammonia? This should be further discussed.

4.      Line 51-52. Is the calorific value of liquid hydrogen very low? This should be checked.

5.      A section should be added to show the future development.

6.      There is no conclusion section in this review, which should be added.

Author Response

Q1. The introduction is too brief. If the reader only reads the introduction, it is hard to think of this as a review of ammonia.

A1. We agree with your comment. We have merged the introduction and second part “Industrial Uses and the Need for Ammonia” (page 2, line 42). We added some sentences as follows: “In addition, biodegradable materials could be used to sources of biofuels and ammonia is expected to be used to one of the biofuels.” (pages 2, lines 37–38).

Q2. The introduction should show the previous reviews on ammonia production, and explain the difference between this paper and the previous reviews and the novelty of this paper.

A2. Thank you for the kind suggestion. We added some sentences as follows: “Several studies have developed sustainable methods to produce ammonia, most of which include green power generation methods, chemical synthesis methods, and sustainable ammonia production using biological methods [58–60]. Various methods of recovering ammonia from food wastes have also been used, and research is progressing toward practical application [61]. Alternatively, we will focus on sustainable ammonia production methods such as biological methods—bacterial and yeast methods—and explore their application potential (Fig. 2).” (pages 4, lines 99–104).

Q3. Biomass is unstable. Is it feasible to use biomass such as food waste to produce ammonia? This should be further discussed.

A3. We agree with you. We have added the sentence “To achieve the sustainable society, we have to solve some problems other than development of biological method such as storage of food waste [122,123].” (page 10, lines 292–293).

Q4. Line 51-52. Is the calorific value of liquid hydrogen very low? This should be checked.

A4. Thank you for the kind suggestion. We have checked the value and changed the sentence from “Hydrogen has low transport efficiency because of its low density and low calorific value per liquid volume.” into “Hydrogen has low transport efficiency because of its low volumetric energy density (3 W h・L-1) and higher calorific value per liquid weight (141.9 MJ/kg) [30].” (page 2, lines 51–52).

Q5. A section should be added to show the future development.

A5. We agree with your assessment. We have added the conclusion section (page 10, lines 282).

Q6. There is no conclusion section in this review, which should be added.

A6. Thank you for the kind suggestion. We have added the conclusion (page 10–12, lines 282–326).

Reviewer 2 Report

Sustainable "green" ammonia production is currently attracting more and more interest. The presented manuscript discusses environmentally friendly approaches alternative to the well-known Haber-Bosch process, mainly focusing on microbial production of ammonia using bacteria and yeast. The content of the review is clear and well-structured; reference list contains relevant and mostly recent publications.

It seems that there are some inaccuracies in the article, so the following notes should be considered:

Fig. 2: Haber – Bosch process, not “Habor bosh process”

Fig. 3: Many crossing and overlapping inscriptions; some of them should be corrected, for instance, “cataboliting genes”, etc. The meaning of the left dark blue box with the oval on top is not clear.  Perhaps it makes sense to add more information summarizing the data on the cassettes encoding nitrogenase expressed in different hosts.

Line 56 – it seems that the words like “higher than” are missing in the phrase “Liquid ammonia contains hydrogen atoms per volume and energy density (MJ L−1) that  are 1.7 and 1.5 times that of liquid hydrogen, respectively [32,33]”.

Lines 151 – 155: “A further attempt was made to reconstitute them by combining 14 nitrogenase-related genes from K. oxytoca into five gene cassettes and cleaving them with a protease from tobacco etch virus after expression. This combination produced E. coli that could grow under anaerobic conditions using only nitrogen molecules as a nitrogen source [76]”. Cleaving of gene cassette by protease seems impossible. The authors of the cited publication (76]) indeed obtained an E. coli strain with reconstruction of the FeFe nitrogenase system, but I did not find any mention of the use of tobacco etch virus protease for this purpose.

Lines 190 – 193: The phrase “In particular, the authors knocked out the codY gene, a regulator of branched-chain amino acids in the B. subtilis metabolic pathway. Furthermore, the authors knocked out bkdB, a lipoamidoacyltransferase, which inhibits the conversion of 2-keto acids to acyl CoA, the raw material for biofuels, thereby promoting biofuel accumulation” is somehow confusing. BkdB knock out does not inhibit the conversion of 2-keto acids to acyl CoA, it inhibits the conversion of branched chain 2-keto acids to their acyl CoA derivatives, thus preventing these 2-keto acids from degrading in an undesirable way. And, in turn, 2-keto acids are decarboxylated by heterologously expressed 2-keto acid decarboxylase to provide biofuel accumulation (in two steps, together with the enzyme alcohol dehydrogenase).

Lines 225: The phrase “With regard to E. coli, many attempts have been made to produce ammonia using  yeast…” is unclear. Probably, you meant “In addition to…”?

Lines 295 -298: The phrase “However, it is active against hydrophobic amino acids such as L-leucine and hydrophilic amino acids such as L-histidine and L-lysine. It is also inactive against aromatic amino acids such as L- histidine, L-tyrosine, and L-phenylalanine” seems to be contradictory and incorrect at least with respect to L-histidine in the second sentence.

Author Response

Q1. Fig. 2: Haber – Bosch process, not “Habor bosh process”

A1. Thank you for your kind comment. I have corrected it (page 4, line 105).

Q2. Fig. 3: Many crossing and overlapping inscriptions; some of them should be corrected, for instance, “cataboliting genes”, etc. The meaning of the left dark blue box with the oval on top is not clear. Perhaps it makes sense to add more information summarizing the data on the cassettes encoding nitrogenase expressed in different hosts.

A2. We agree with your assessment. I have changed the terms and the contents of the slide (page 5, line 113 and page 11, line 309).

Q3. Line 56 – it seems that the words like “higher than” are missing in the phrase “Liquid ammonia contains hydrogen atoms per volume and energy density (MJ L−1) that are 1.7 and 1.5 times that of liquid hydrogen, respectively [32,33]”.

A3. Thank you for your suggestion. I have modified the sentence “Liquid ammonia contains hydrogen atoms per volume and energy density (MJ L−1) that are 1.7 and 1.5 times that of liquid hydrogen, respectively [32,33]”. Into “Liquid ammonia contains hydrogen atoms per volume and energy density (MJ L−1) that are 1.7 and 1.5 times higher than that of liquid hydrogen, respectively [32,33]” (page 3, lines 54–55).

Q4. Lines 151 – 155: “A further attempt was made to reconstitute them by combining 14 nitrogenase-related genes from K. oxytoca into five gene cassettes and cleaving them with a protease from tobacco etch virus after expression. This combination produced E. coli that could grow under anaerobic conditions using only nitrogen molecules as a nitrogen source [76]”. Cleaving of gene cassette by protease seems impossible. The authors of the cited publication (76]) indeed obtained an E. coli strain with reconstruction of the FeFe nitrogenase system, but I did not find any mention of the use of tobacco etch virus protease for this purpose.

A4. You raise an important question. I have cited appropriate reference article (page 6, lines 140–143).

Q5. Lines 190 – 193: The phrase “In particular, the authors knocked out the codY gene, a regulator of branched-chain amino acids in the B. subtilis metabolic pathway. Furthermore, the authors knocked out bkdB, a lipoamidoacyltransferase, which inhibits the conversion of 2-keto acids to acyl CoA, the raw material for biofuels, thereby promoting biofuel accumulation” is somehow confusing. BkdB knock out does not inhibit the conversion of 2-keto acids to acyl CoA, it inhibits the conversion of branched chain 2-keto acids to their acyl CoA derivatives, thus preventing these 2-keto acids from degrading in an undesirable way. And, in turn, 2-keto acids are decarboxylated by heterologously expressed 2-keto acid decarboxylase to provide biofuel accumulation (in two steps, together with the enzyme alcohol dehydrogenase).

A5. Thank you for this suggestion. I have modified these sentences “Furthermore, the authors knocked out bkdB, a lipoamidoacyltransferase, which inhibits the conversion of 2-keto acids to acyl CoA, the raw material for biofuels, thereby promoting biofuel accumulation” from “Furthermore, the authors knocked out bkdB, a lipoamidoacyltransferase, Then, 2-keto acids are decarboxylated by heterologously expressed 2-keto acid decarboxylase to provide biofuel accumulation.”  (page 7, lines 176–178).

Q6. Lines 225: The phrase “With regard to E. coli, many attempts have been made to produce ammonia using  yeast…” is unclear. Probably, you meant “In addition to…”?

A6. In accordance with the reviewer's comment, we have changed this to In addition to… (page 8, line 207).

Q7. Lines 295 -298: The phrase “However, it is active against hydrophobic amino acids such as L-leucine and hydrophilic amino acids such as L-histidine and L-lysine. It is also inactive against aromatic amino acids such as L- histidine, L-tyrosine, and L-phenylalanine” seems to be contradictory and incorrect at least with respect to L-histidine in the second sentence.

A7. Accordingly, we have changed the sentence “However, it is active against hydrophobic amino acids such as L-leucine and hydrophilic amino acids such as L-histidine and L-lysine. It is also inactive against aromatic amino acids such as L- histidine, L-tyrosine, and L-phenylalanine” to “However, it is active against hydrophobic amino acids such as L-leucine and hydrophilic amino acids such as L-histidine and L-lysine. It is also inactive against aromatic amino acids such as L-tyrosine, and L-phenylalanine” (page 9, lines 268–269).

Reviewer 3 Report

This paper summarizes the recent progress in utilizing biological methods (especially bacteria and yeast) for sustainable ammonia production. The review paper is well-organized, and the context is easy to follow. However, in my opinion, this paper does not qualify as publishable content in its current form, especially with the confusing figures and poor schematic illustrations. The authors are suggested to carefully revise the paper before it can be considered for publication.

Comments:

1.      The last sentence of section 1 “Introduction” does not make sense. “The biodegradable materials were widely used” is not the reason of “their products do not have a negative effect on the environment”. As mentioned in the paper, fossil fuels were also widely used, but there are negative effects of fossil fuels on the environment. Please re-write this sentence to make it reasonable.

2.      Figure 1. I am confused about the figure. In my opinion, these energy sources are parallel, and wind/polar energy or food waste are not converted from fossil fuels.

3.      The “created with Biorender” in figure 1 should be annotated in the figure legend instead of in the figure. This also applies to other figures in the manuscript. Please revise accordingly.

4.      Figure 2 has the same problem as figure 1. What does the arrow indicate? Are the biological processes derived from the chemical process? If the authors are aiming to compare the two methods, the “arrow” is not appropriate.

5.      Figure 3. There should be different panels for the two schemes (also please annotate in the legend). The text is overlapping with the schemes which makes the figure hard to read and it is unpublishable. The authors should follow the instruction from the journal for how to draw appropriate figures for their paper.

6. In Section 4, more references should be added to support this statement. Here are a few good examples: https://doi.org/10.1016/j.biotechadv.2018.10.009; https://doi.org/10.1016/j.tibtech.2019.11.007; and https://doi.org/10.1039/D0CS00155D.

7.      There are several font errors in the section 4, for example, Line 12 of the fourth paragraph in section 4. The font of this sentence is obviously larger than the other context. The authors should carefully revise and proofread their manuscript before submitting it for publication.

8.      A table summarizing the specific strategies used in biological processes is needed. The current table is too general. The addition of such a table would highlight the focus of this paper, which is using biological methods for sustainable ammonia production. 

Author Response

Q1. The last sentence of section 1 “Introduction” does not make sense. “The biodegradable materials were widely used” is not the reason of “their products do not have a negative effect on the environment”. As mentioned in the paper, fossil fuels were also widely used, but there are negative effects of fossil fuels on the environment. Please re-write this sentence to make it reasonable.

A1. Thank you for this kind suggestion. I changed the sentence of the end of introduction from “In addition, biodegradable materials have been widely used; thus, their products do not have a negative effect on the environment (Fig. 1) [14].” into ” In addition, biodegradable materials have been widely produced. If they are used for the production of carbon neutral fuel, their consumption does not have a negative effect on the environment (Fig. 1) [7].” (page 2, lines 37–38).

Q2. Figure 1. I am confused about the figure. In my opinion, these energy sources are parallel, and wind/polar energy or food waste are not converted from fossil fuels.

A2. Thank you for your kind opinion. I have changed the direction of the arrow in Fig. 1 to show the contrast between fossil fuel, and food waste or green energy (page 2).

Q3. The “created with Biorender” in figure 1 should be annotated in the figure legend instead of in the figure. This also applies to other figures in the manuscript. Please revise accordingly.

A3. We agree you. I have moved these sentences into figure legends (page 2, line 40, page 4, line 105, page 5, line 113 and page 11, line 309).

Q4. Figure 2 has the same problem as figure 1. What does the arrow indicate? Are the biological processes derived from the chemical process? If the authors are aiming to compare the two methods, the “arrow” is not appropriate.

A4. Thank you for your suggestion. I have changed the direction of the arrow in Fig. 2 to show the contrast between Habor–Bosch process, and biological process (page 4, line 105).

Q5. Figure 3. There should be different panels for the two schemes (also please annotate in the legend). The text is overlapping with the schemes which makes the figure hard to read and it is unpublishable. The authors should follow the instruction from the journal for how to draw appropriate figures for their paper.

A5. Agreed. I have separated Fig. 3 into two parts, Fig. 3 and Fig4 (page 5, lines 113 and page 11, lines 309).

Q6. In Section 4, more references should be added to support this statement. Here are a few good examples: https://doi.org/10.1016/j.biotechadv.2018.10.009; https://doi.org/10.1016/j.tibtech.2019.11.007; and https://doi.org/10.1039/D0CS00155D.

A6. Thank you for providing these insights. I have added the sentence “Microorganisms can degrade and synthesize a wide variety of compounds by applying engineering method like directed evolution or genome editing [63–66].” with some papers (page 5, line 108–109).

Q7. There are several font errors in the section 4, for example, Line 12 of the fourth paragraph in section 4. The font of this sentence is obviously larger than the other context. The authors should carefully revise and proofread their manuscript before submitting it for publication.

A7. We agree with your assessment. I have changed the font (Times new roman) and the size (11 point) of this manuscript.

Q8. A table summarizing the specific strategies used in biological processes is needed. The current table is too general. The addition of such a table would highlight the focus of this paper, which is using biological methods for sustainable ammonia production.

A8. Thank you for your kind suggestion. I have added a new table (page 10, line 290).

Reviewer 4 Report

Watanabe et al present a comprehensive review about microbial NH3 production using particularly food waste as a resource. The selection of literature is well done. However, my concerns consider the manner the manuscript is structured:

1) The authors outline the pros applying microbes but they do not outline that about 20% of nitrogen would be used for biomass formation (given the typical composition of cells). I would recommend to consider a likewise statement somewhere.

2) The authors do not comment on potential conflicts draining food residuals away for NH3 production instead of using them e.g. in anaerobic digestion plants etc. Again, a remark would be helpful complementing the review.

3) Chapter 4 would greatly benefit from considering subtitles, i.e. introducing a structure that guides ideas of the reader. For instance, distinguish between homologeous and heterologous approaches, growth/non-growth coupled production, amplification of internal production pathways versus interruption of further internal use etc. Right now, chapter 4 is a little bit lengthy for reading...

4) The last paragraph 'The use of metabolically optimized ...' comes as a surprise. Apparently, authors wanted to write a take-home message but this should be done in a separate Conclusion. Furthermore, the reader should be prepared for said conclusion. Right now, the link is missing.

5) The reviewer would have appreciated to get a collection of basic microbial kinetics aforehand. I.e. microbial uptake affinities, growth inhibition constants, export rates etc. Such values are important to quantify the cellular potential.   

Author Response

Q1. The authors outline the pros applying microbes but they do not outline that about 20% of nitrogen would be used for biomass formation (given the typical composition of cells). I would recommend to consider a likewise statement somewhere.

A1. Thank you for your kind suggestion. I have changed the sentence “Also, microorganisms like E. coli and yeast contain nitrogen in its body about 20% as a protein [129]. The nitrogen in cell comes from assimilation of amino acid or protein in medium. Therefore, the amount of produced nitrogen must be larger than input nitrogen for effective biofuel production. To solve this problem, cell surface engineered yeast could be repeatedly used for production because the enzyme on the yeast cell surface is stable [130].

” in conclusion part (page 10, lines 299–303).

Q2. The authors do not comment on potential conflicts draining food residuals away for NH3 production instead of using them e.g. in anaerobic digestion plants etc. Again, a remark would be helpful complementing the review.

A2. You raise an important question. I have added the sentence about anaerobic digestion in conclusion part “In addition to studies in this review, for example, anaerobic culture is also one of the methods currently in practical use for energy production from food wastes [120]. These anaerobic cultures usually do not use genetic engineering methods, so that this method has room for optimization by using genetic engineering introduced in this review.  ”(Page10, lines 285–288).

Q3. Chapter 4 would greatly benefit from considering subtitles, i.e. introducing a structure that guides ideas of the reader. For instance, distinguish between homologeous and heterologous approaches, growth/non-growth coupled production, amplification of internal production pathways versus interruption of further internal use etc. Right now, chapter 4 is a little bit lengthy for reading...

A3. Thank you for this suggestion. I have added the title “2.1. Bioengineering of nitrogen fixation and metabolic engineering for ammonia production” in page 5, line 114 and “2.2.          Ammonia production from food waste by using bioengineering methods” page 7, line 170.

Q4. The last paragraph 'The use of metabolically optimized ...' comes as a surprise. Apparently, authors wanted to write a take-home message but this should be done in a separate Conclusion. Furthermore, the reader should be prepared for said conclusion. Right now, the link is missing.

A4. You have raised an important point. We have moved the sentence into conclusion in page 11, lines 311–315.

Q5. The reviewer would have appreciated to get a collection of basic microbial kinetics aforehand. i.e. microbial uptake affinities, growth inhibition constants, export rates etc. Such values are important to quantify the cellular potential.

A5. We agree with your assessment. We have added the sentence “Recently, some studies attempt to estimate the metabolism of microorganisms so that it might be possible to calculate the potential of microorganisms to produce ammonia efficiently in the future [117–119].” in page 10–11, lines 303–305.

Round 2

Reviewer 1 Report

The revised manuscript can be accepted.

Reviewer 3 Report

All my comments have been addressed. One additional comment is about the figure quality. The authors should use/upload figures with high resolution and dpi. The current one is not suitable for publication. Other than that, I do not have any further comments. 

Reviewer 4 Report

I accept the author's changes